

# Evaluation of the global-blockage effect on power performance through simulations and measurements

Alessandro Sebastiani[1], Alfredo Peña[1], Niels Troldborg[1], and Alexander Meyer Forsting[1]

[1]DTU Wind Energy, Frederiksborgvej 399 4000, Roskilde, Denmark

**Correspondence:** Alessandro Sebastiani (aseb@dtu.dk)

**Abstract.**

Blockage effects due to the interaction of five wind turbines in a row are investigated through both Reynolds-averaged Navier-Stokes simulations and site measurements. Since power performance tests are often carried out at sites consisting of several turbines in a row, the objective of this study is to evaluate whether the power performance of the five turbines differs from that of an isolated turbine. A number of simulations are performed, in which we vary the turbine inter-spacing (1.8, 2 and 3 rotor diameters) and the inflow angle between the incoming wind and the orthogonal line to the row (from 0° to 45°). Different values of the free-stream velocity are considered to cover a broad wind speed range of the power curve. Numerical results show consistent power deviations for all the five turbines when compared to the isolated case. The amplitude of these deviations depends on the location of the turbine within the row, the inflow angle, the inter-spacing and the power curve region of operation. We show that the power variations do not cancel out when averaging over a large inflow sector (from $-45°$ to $+45°$) and find an increase in the power output of up to $+1\%$ when compared to the isolated case. We simulate power performance 'measurements' with both a virtual mast and nacelle-mounted lidar and find a combination of power output increase and upstream velocity reduction, which causes an increase of $+4\%$ of the power coefficient. We also use measurements from a real site consisting of a row of five wind turbines to validate the numerical results. From the analysis of the measurements, we also show that the power performance is impacted by the neighbouring turbines. Compared to when the inflow is perpendicular to the row, the power output varies of $+1.8\%$ and $-1.8\%$ when the turbine is the most downwind and upwind of the line, respectively.

## 1 Introduction

It is well known that the performance of a wind turbine is highly affected by the wakes of upstream turbines (Crespo et al., 1999; Barthelmie et al., 2009; Göçmen et al., 2016; Sebastiani et al., 2021). Also well known is the blockage effect, which is the reduction of the velocity upstream of the turbine, due to the presence of the turbine itself (Medici et al., 2011; Meyer Forsting, 2017). Recently, the global-blockage effect has started to draw attention within the wind energy community. This



is also characterized by a velocity decrease but upstream of a wind farm or cluster of wind turbines, due to the presence of

the wind farm itself (Bleeg et al., 2018). In the latter study, the global-blockage effect was quantified by comparing wind speed measurements collected from meteorological masts before and after the operation of wind farms. Additionally, they showed, using Reynolds-averaged Navier-Stokes (RANS) simulations, that the velocity reduction upstream of wind farms causes the energy production of each of the turbines in the front row of the wind farm to be different from that of the same turbine in isolation. The velocity reduction upstream of a wind farm made up of several rows was also shown by Segalini and

Dahlberg (2020) using wind tunnel experiments. Schneemann et al. (2021) showed how the global-blockage effect relates to the atmospheric stability for the case of an offshore wind farm. They used a long-range Doppler scanning lidar to measure the wind speed upstream of the wind farm and showed global blockage only under stable atmospheric conditions.

The global-blockage effect is not only due to the superposition of the induction from the single turbines, but also the result of the interaction between the wind farm as a whole and the atmospheric boundary layer (Allaerts and Meyers, 2017; Porté-Agel

et al., 2020). This interaction generates an upstream reverse pressure gradient and thus a wind speed reduction. Meyer Forsting et al. (2021) tested several induction models based on the linear summation of the blockage from contributions of the single turbines and showed good agreement with RANS simulations without modeling the atmospheric boundary layer. In this work, with the term global blockage, we refer to all the alterations of the flow field (out of the wind-turbine wakes) caused by the presence of a number of wind turbines, which would not occur for the case of an isolated turbine. Furthermore, some

studies showed global blockage for a single row of turbines, where turbines are affected by those besides them rather than by downstream turbines (McTavish et al., 2015; Meyer Forsting et al., 2017b). The power output of three wind turbines aligned perpendicularly to the wind was shown to be higher than that of an isolated turbine by means of both wind tunnel studies and simulations with the free-vortex code GENUVP (McTavish et al., 2015). They explained that the power increase is a consequence of in-field blockage occurring between adjacent turbines, which results in a region of relative increased wind

speeds that extends up to three rotor diameters (D) upstream of the row. For the case with a spacing of 2D, they found an increase of power output of the order of 3% compared to the isolated turbine.

According to the IEC standard (IEC, 2017), power performance testing can be performed on a turbine within a row of turbines by considering a wind sector within the direction perpendicular to the row (±50° when the turbine inter-spacing is 2D). Within these inflow conditions, turbines are assumed to be unaffected by neighbouring turbine wakes and the measured

power curve is assumed to be valid for the case of an isolated turbine. The study of McTavish et al. (2015) was perhaps the first that questioned these assumptions. Meyer Forsting et al. (2017b) analyzed the power production of turbines in a row by using both RANS simulations and a simple inviscid vortex ring model with wake expansion. They considered a row of five turbines with a 3D turbine spacing, a wind speed of 8 m/s (in the middle between cut-in and rated values), and wind directions of +0°, +15°, +30°and +45°relative to the orthogonal line to the row. Results showed a difference in the power output when comparing

each of the turbines in the row to the isolated case, which depended on the inflow angle and the location of the turbine in the row. The largest difference (2%) was found for the turbine on the row edge for an inflow angle of 45°.

Ideally, a power curve relates the power output of the turbine with the wind speed that would be measured at the turbine's location without the turbine actually being there. The IEC standard assumes that blockage is negligible already at 2D in front of





the turbine and suggests to measure either the hub-height wind speed or the rotor-equivalent wind speed (Wagner et al., 2011) in front of the turbine at a distance between 2D and 4D. At these upstream distances, global blockage influences the flowfield and the power production (Meyer Forsting et al., 2017b; Bleeg et al., 2018; Segalini and Dahlberg, 2020), indicating that standard power performance tests, normally carried out on turbines at sites with at least a row of turbines might be affected by global blockage. In this work, we use a similar numerical experiment to that of Meyer Forsting et al. (2017b) to further investigate this issue. We analyze the power output of five wind turbines in a row and investigate the difference to their production in isolation. We extend the numerical work of Meyer Forsting et al. (2017b) by extracting velocities in front of the turbines using virtual met masts and nacelle-mounted lidars to further analyze the relation between global blockage and power performance measurements. Additionally, we analyze if and how deviations in the power output between the row and the isolated case are affected within a broad range of free-stream velocities and turbine inter-spacings. The inflow velocities cover a number of regions of the power curve from cut-in to rated, while the turbine inter-spacings represent typical values used at test sites. Further, this work includes the analysis of measurements from a real site consisting of a row of five wind turbines. This is the first time that the global-blockage effect for a single row of turbines is investigated using both simulations and measurements, which comply with the IEC standard for power curve measurements.

This paper is organized as follows. In Sect. 2, the numerical setup and the available measurements are introduced. In 2.2.1, possible numerical biases are analyzed. The numerical results are reported in Sect. 3; the global-blockage effect on the power output is shown in Sect. 3.1, variations of the flow field around the row are shown in Sect. 3.2 and effects on power performance measurements are analyzed in Sect. 3.3. The analysis of the measurements is described in Sect. 4. In Sect. 4.1, we explain how the measurements are filtered to assure compliance with the numerical setup. In Sect. 4.2 the power variations observed in the measurements are compared with those of the simulations. Finally, a discussion and conclusions are presented in Sect. 5 and 6, respectively.

## 2 Methodology

### 2.1 Problem definition

The numerical setup consists of five turbines aligned in a row perpendicular to the prevailing wind, similar to power performance test sites. In addition to the case of the wind approaching perpendicular to the row, $\theta = 0°$, inflow angles between $5°$ and $45°$ are considered, as shown in figure 1. The modelled turbine is the NREL 5-MW with a diameter of 126 m (Jonkman et al., 2009), but any other turbine type could have been used as blockage is largely turbine design independent Meyer Forsting (2017); Meyer Forsting et al. (2021). The effect of the turbine spacing ($L$) is evaluated by considering three different values: $L$ = 1.8D, 2D, and 3D. The 1.8D case is tested to evaluate whether the global-blockage effect changes dramatically for a spacing lower than 2D, which is the lowest value currently accepted by the IEC standard. To highlight the effects of the rotors on the power output, the inflow is simplified as much as possible. Therefore, the inflow is uniform without turbulence and assumed as time invariant. Virtual measurements from meteorological towers are simulated by extracting point-wise velocity values in front of the rotors at hub height and at 2D, 2.5D and 3D upstream, which are the distances prescribed in the IEC standard. Lidar





measurements are simulated with a two-beam pulsed lidar mounted on the nacelle and pointing upstream with a half-opening angle of 15°. The lidar is characterized by a range-gate length of 38.4 m and a full width half maximum (FWHM) of 24.75 m. More details about the lidar simulator can be found in Meyer Forsting et al. (2017a). As it can be seen in figure 1, the mast
measurements are taken at fixed locations, while the nacelle-mounted lidars yaw together with the rotors and their point of measurement changes with $\theta$.

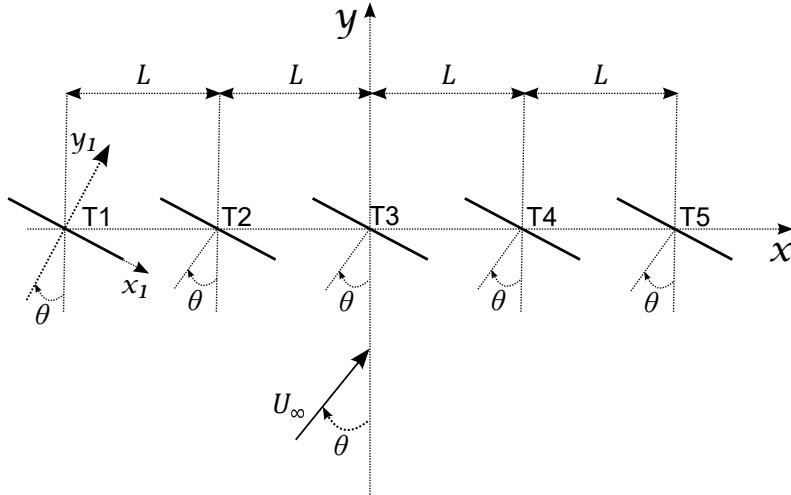

**Figure 1.** Schematics of the wind turbine rotors (T1–T5) in the numerical setup. Clockwise relative directions of $\theta$ are positive.

## 2.2   Computational method

The numerical setup adopted here is the same as used and described in detail by Meyer Forsting et al. (2017b) so here we will only briefly describe the simulation setup. All simulations are performed using the in-house incompressible finite volume flow
solver EllipSys3D (Michelsen, 1992, 1994; Sørensen, 1995). The simulations are carried out as steady state Reynolds Average Navier-Stokes (RANS) using the $k - \omega$ SST turbulence model by Menter (1994). The numerical domain is an ellipse shaped cylinder with $(L_x, L_y, L_z) = (95D, 84D, 25D)$, where $L_x$ and $L_y$ denote the major and minor axis of the ellipse and $L_z$ is the height of the cylinder. The turbines are placed as shown in Fig. 1 with T3 located in the centre of the domain. In the vicinity of the turbines, the grid cells are cubic with a side length of $D/32$ within an inner box of dimensions $(15D, 4D, 2D)$. From there,
the mesh grows hyperbolically outwards. The turbines are modelled as actuator disks (Réthoré and Sørensen, 2012; Troldborg et al., 2015) using the airfoil and blade data from the NREL 5-MW turbine (Jonkman et al., 2009). In contrast to Meyer Forsting et al. (2017b), who prescribed a constant rotational speed and blade pitch angle, we instead use a controller that set these based on the velocity averaged over the rotor area at the rotor position (Van Der Laan et al., 2015).

The accuracy of the CFD model (numerical setup and actuator disk) over the wind turbine induction zone was validated
using measurements from three lidars (Meyer Forsting et al., 2017).





### 2.2.1 Sensitivity to numerical domain and turbine location

As we need to assess the difference in both inflow and power output between a row of turbines and an isolated turbine, we need to verify that the difference between the two cases is only due to the number of turbines without being affected by numerical bias. Meyer Forsting and Troldborg (2015) already showed that this numerical setup guarantees results free of tunnel blockage
due to either grid resolution or domain size.

Here, we further simulate an isolated turbine placed at the location of turbine 5 and these results are compared with the reference case, i.e. an isolated turbine placed at T3, for $U_\infty = 8$ m/s and $\theta = 0°, 30°$ and $45°$. The difference in power output is found to be negligible compared to the deviations caused by the whole row (we show these deviations in Sect. 3.1). Specifically, for $\theta = 45°$, when the turbine is placed at T5, the power output is only 0.15% higher than when it is placed at T3.

The results might also be biased due to numerical sensitivity to the inflow angle, as the effective grid resolution changes when the flow is aligned or misaligned with the grid direction. Even though the grid refinement is unchanged, these variations of the effective resolution affect the power output of the isolated turbine, which changes with $\theta$ while it should be dependent on $U_\infty$ alone. However, this effect causes only small variations of power. The difference with the case $\theta = 0°$ is of 0.02% for $\theta = 30°$ and of 0.15% for $\theta = 45°$.

Simulations are also performed with a staggered configuration, where the same layout of figure 1 is achieved not by yawing the rotors, but by moving the turbines along the $y$ direction, as it is shown in figure 2 . In this way, the main flow direction is aligned with the grid direction regardless of the inflow angle. In the staggered configuration, the equispaced box mesh in the centre of the domain is enlarged ($y = -4\mathrm{D}$ to $y = +4\mathrm{D}$), as T1 and T5 would otherwise be out of the refined area. Results show a much larger dependency of the power output on the turbine location, with variations for the power of the isolated turbine
when it is placed at different locations. This is probably due to differences in the fraction of the wake that rests inside the refined mesh region, causing variations for the induction of the single rotors. The power output of the isolated turbine decreases 1.2% when the turbine is moved from T3 to T1, while it increases 0.71% when moved from T3 to T5. These results suggest that the staggered configuration should be avoided for studies that require high accuracy and, thus, not used in this study.

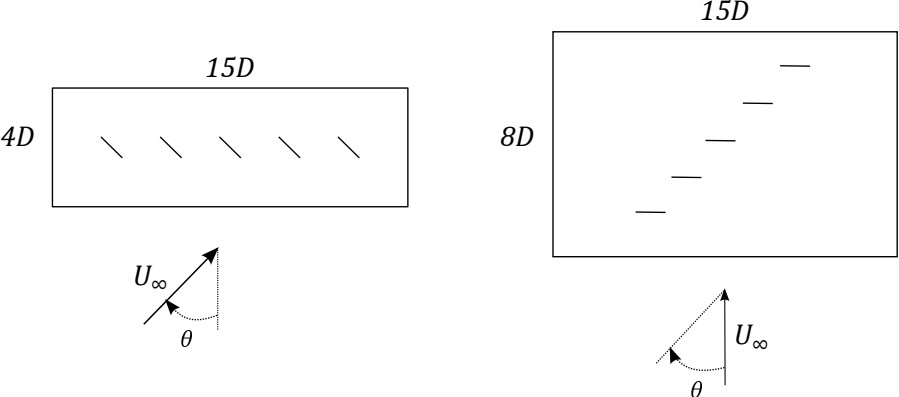

**Figure 2.** Standard and staggered configurations with boundaries of the refined area.





### 2.3 Measurements

Measurements are available for a period of approximately 21 months from a site, which consists of five turbines aligned perpendicularly to the predominant wind direction. The area is flat and the surface characteristics within the analyzed directions are the same for each of the turbines in the row (and rather homogeneous). The name of the site can not be disclosed due to proprietary reasons, but the layout is very similar to that in figure 1. The available dataset comprises the operational data from a turbine on one edge of the row (T1) together with measurements from 'its power-performance' meteorological mast

and a ground-based wind lidar aligned with the turbine along the predominant wind direction at distances of 2.3D and 2.5D, respectively. The lidar is a WindCube WLS7 from Vaisala Leosphere. Additionally, the data include the operational status of the turbines T3–T5 (T2 operation and status is unknown). The five turbines are placed with a mutual distance $L = 2.3D$, with D being the diameter of T1. Although we do not know specifics on the turbines standing on the other four positions, considering the size of modern wind turbines, the spacing is likely to be lower than 3D when normalized with the rotor diameters of the

other turbines at the site.

  Data from T1 is used to validate the numerical results. If the asymmetry due to wake rotation is neglected, the turbine can represent either turbine T1 or T5 from the simulations, as it is either the most upwind or downwind turbine of the row for $\theta > 0°$ and $\theta < 0°$, respectively.

  Measurements from both the turbine and the mast are sampled at 35 Hz, while the wind lidar provides measurements at 11

different heights every 4 s, covering a vertical distance from $-0.4D$ to $+0.85D$ relative to the wind turbine hub height. The analysis is performed by considering 10-min means for all examined variables.

### 3 Numerical results

### 3.1 Power output

  In figure 3, the power output $P$ of the five turbines is normalized by that of the isolated turbine $P_{ref}$ under the same inflow

conditions. The normalized power varies with the free-stream velocity $U_\infty$ for different values of $\theta$. At $U_\infty \approx 8$ m/s, the turbine is within the region of the power curve where the turbine controller keeps a constant tip speed ratio and an optimal power output, i.e. a constant power coefficient ($C_P$) and thrust coefficient ($C_T$). We also show results for 7 and 11 m/s as they are the first two integer values out of this region (Jonkman et al., 2009). The highest variation from the reference case is found for the side turbines (T1 and T5) with an inflow angle $\theta = 45°$. Although not shown, the difference between the power output

of the five turbines and the isolated turbine decreases for 12 m/s (above the $C_P$-constant interval), while it increases for 7 m/s. The power output increases for all the five turbines when $\theta = 0°$, with the highest gain, when compared to the reference case, for T3 and $U_\infty = 7$ m/s reaching nearly 2%.

  Figure 4 shows the normalized power output for cases with the same free-stream velocity (8 m/s) and a number of turbine inter-spacings (1.8D, 2D and 3D). The normalized power varies with turbine inter-spacing for all the five turbines and decreases

the larger the turbine inter-spacing. For the largest turbine inter-spacing (3D), the normalized power is still larger than 2% for

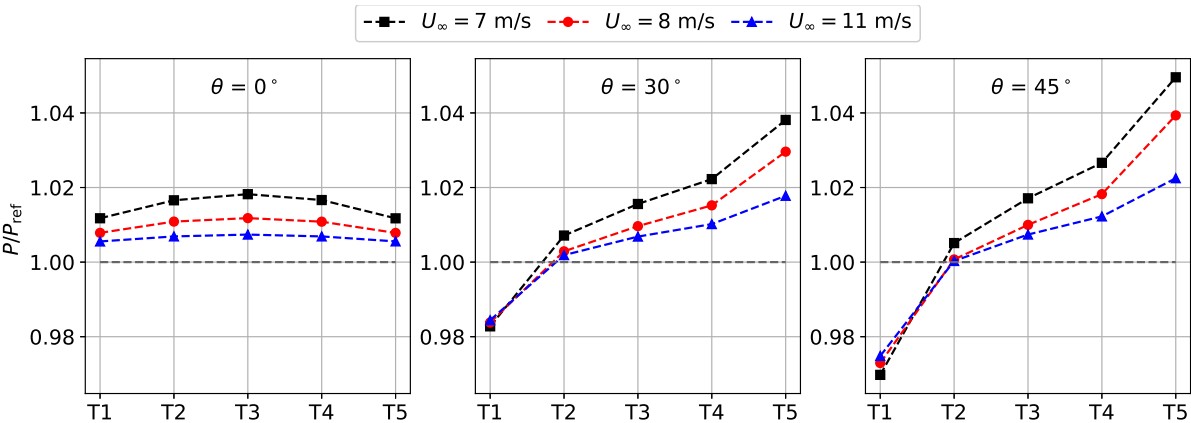

**Figure 3.** Power output of the five turbines normalized by that of an isolated turbine (placed at T3) for the cases with an inter-spacing of 2D predicted by RANS-CFD.

the side turbines when $\theta = 30°$ and $45°$. The reduction of the turbine inter-spacing from 2D to 1.8D results in small variations of the power output; the highest variation (0.75%) is for turbine T5 when $\theta = 0°$.

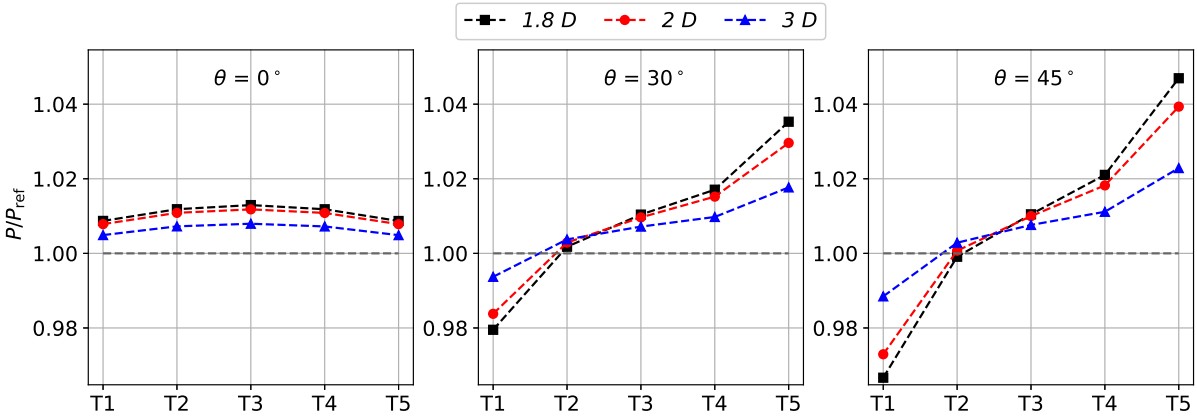

**Figure 4.** Power output of the five turbines normalized by that of an isolated turbine (placed at T3) for $U_\infty = 8$ m/s and a number of $\theta$ values and turbine inter-spacings.

### 3.2 Global blockage and induced velocities

The higher power output of the five turbines relative to that of the isolated case cannot be explained with upstream velocity
measurements. The upstream induction on the row of turbines is higher than that of the single turbine, so that there is a higher velocity reduction in front of the rotors, as expected because of the global-blockage effect. This is shown in figure 5, where





the vertical velocity profile in front of T1 and T3 is compared to that of an isolated turbine for $U_\infty = 8$ m/s and $\theta = 0°$. Lower velocities correspond to higher power production, with T3 producing the most despite of the lowest incoming wind speed at both 2D and 1D. It is only very close to the rotor (closer than 0.2D) that the incoming wind speed in front of T3 is higher than
in the isolated case. Although not shown, the same trend is found for all values of $U_\infty$ and $\theta$.

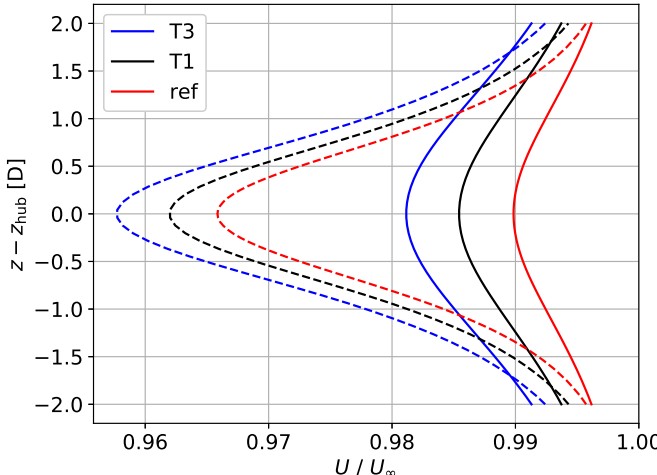

**Figure 5.** Upstream vertical velocity profiles extracted from RANS-CFD at 1D (*dashed lines*) and 2D (*continuous lines*) of the isolated case, T1 and T3 for $U_\infty = 8$ m/s and $\theta = 0°$.

Meyer Forsting et al. (2017b) already showed that these counter-intuitive power deviations of the turbines on the row relate to the downstream induced velocity caused by the neighbouring turbines. Particularly, a positive downstream induced velocity results in faster advection of the wake and lower induction upstream of the turbine. The 'local' blockage at the rotor is thus lower compared to the isolated case, which results in higher power output. Likewise, a negative downstream induced velocity
results in lower power output compared to the isolated case.

Figure 6 shows the velocity induced by the isolated turbine at T3 along the rotor axis at the locations T2 and T4 (but without other rotors than T3) for $\theta = 45°$ and $U_\infty =7, 8$ and $11$ m/s. For $-1.3 \lesssim y_i \lesssim 1.5$, the induction is positive along $y_4$ and negative along $y_2$. This explains the results in figure 3, where the downstream turbines (T4 and T5) produce more than the upstream ones (T1 and T2). Additionally, the magnitude of the induction decreases the higher the wind speed, also in agreement with
the results in figure 3, where the power variation decreases for higher wind speeds. Furthermore, as shown in figure 7, the magnitude of the induced velocities varies with the turbine inter-spacing so that stronger inductions are observed for an inter-spacing of 2D compared to those of the 3D case, which is in agreement with the power variations in figure 4. The variation of induced velocities with both turbine inter-spacing and wind speed further confirms the relation between downstream induced velocities and power variations.

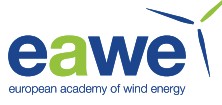


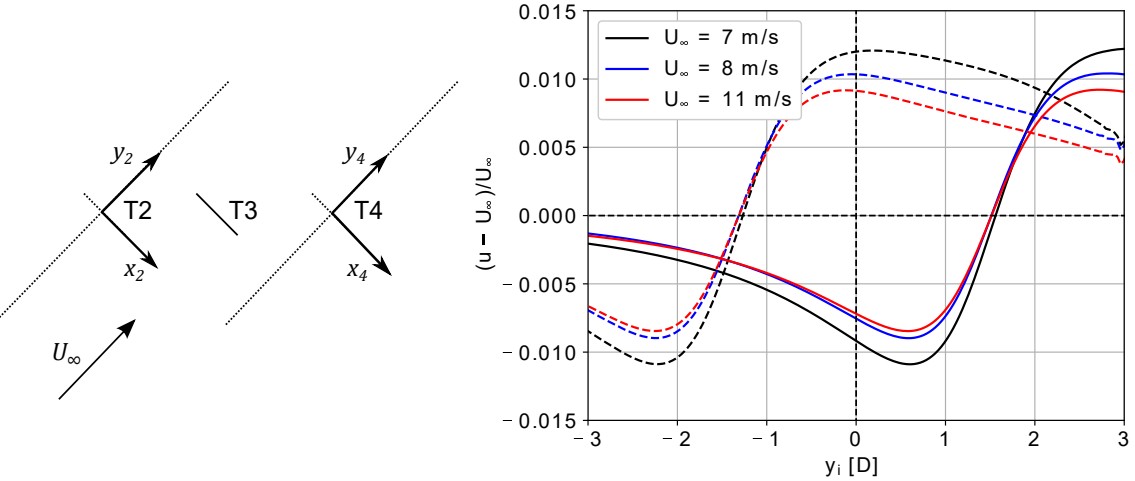

**Figure 6.** Velocities induced by the central turbine (T3) on the rotor axis of T2 (*solid line*) and T4 (*dashed line*), for different values of $U_\infty$, turbine inter-spacing of 2D and $\theta = 45°$.

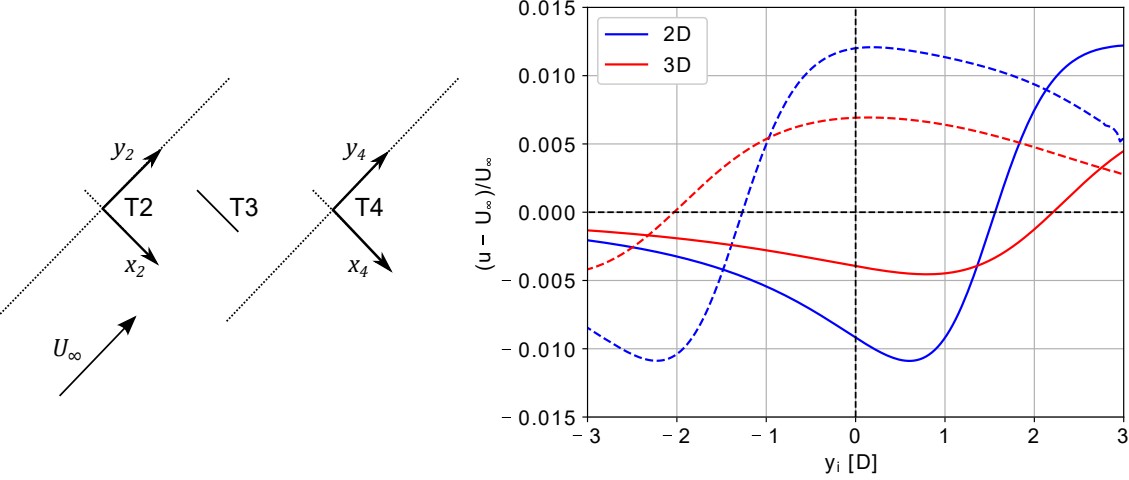

**Figure 7.** Velocities induced by the central turbine (T3) on the rotor axis of T2 (*solid line*) and T4 (*dashed line*), for $U_\infty = 7$ m/s, $\theta = 45°$ and different turbine inter-spacings.





## 3.3 Effects on power performance measurements

From the previous results, one might expect biases in power performance measurements carried out for non-isolated turbines. Particularly, we would like to quantify whether the effects shown for specific $\theta$ values in figures 3 and 4 cancel out when averaging over an inflow sector typical for power performance measurements.

A series of simulations are performed for both the reference case and the turbine row with an inter-spacing of 2D for a number of $U_\infty$ and $\theta$ values. The free-stream velocity varies from 7 to 11 m/s with a step of 1 m/s, while $\theta$ varies from $-45°$ to $+45°$ with a step of $5°$. A normal distribution $N_\theta(\mu_\theta, \sigma_\theta^2)$ (with $\mu_\theta$ and $\sigma_\theta$ as mean and standard deviation) is assumed for the wind direction and the mean power output is calculated for each free-stream velocity as $\bar{P} = \int P(U_\infty, \theta)\, N_\theta(\mu_\theta, \sigma_\theta^2)\, d\theta$. The effect of averaging over the whole inflow sector is shown in figure 8 for a distribution given by $\mu_\theta = 0°$ and $\sigma_\theta = 41°$. As illustrated, there is a difference with respect to the reference case; the five turbines in the row produce more than in isolation for all values of $U_\infty$. Since $N_\theta(\mu_\theta, \sigma_\theta^2)$ is symmetric and centered in $\theta = 0°$, the power output of T5 is exactly the same as T1 and the same applies to T2 and T4. The central turbine T3 shows the largest increase of power relative to the reference case, with a power gain higher than $1\%$ for most wind speeds. Furthermore, the highest and lowest power variations are observed for $U_\infty = 7$ and 11 m/s, respectively. The power variations are nearly constant for free-stream velocities within the range 8–10 m/s. These results further confirm that the global-blockage-related power variations depend on the power curve region of operation of the wind turbines; they are $C_T$-dependent and consequently steady in the constant-$C_P$ region of the power curve, while they decrease for $U_\infty$ closer to the rated wind speed and increase for $U_\infty$ closer to the cut-in value.

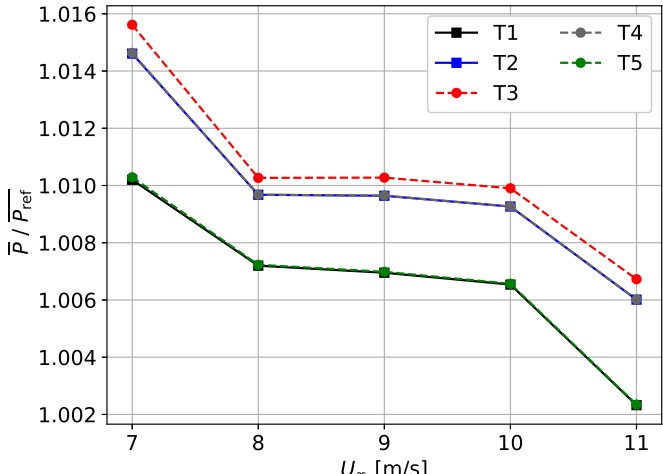

**Figure 8.** Power output of the five turbines in the row averaged over an inflow sector and normalized by the average power output of the isolated turbine for several values of $U_\infty$ and a normal distribution $N_\theta(\mu_\theta = 0°, \sigma_\theta = 41°)$ for $\theta$.

Due to the increase in power and reduction of the upstream wind speed, the differences in power coefficient $C_P$ compared to the isolated case are higher than those of the power output. Results for $C_P$ are shown in figure 9 for the same case of figure





8. The estimated free-stream velocity $U_\infty$ is extracted at hub height and 2.5D upstream of the rotor by both the virtual met

mast and the virtual nacelle-mounted lidar. $C_P$s for the turbines of the row are up to 4% higher than that of the reference when

measuring with a mast and higher when measuring with the nacelle lidar. This is due to the masts measuring at fixed locations,

while the nacelle-mounted lidars yaw together with the turbine. The volume-averaging effect of the lidar is considered as

negligible due to the nearly uniform velocity within the probe volume.

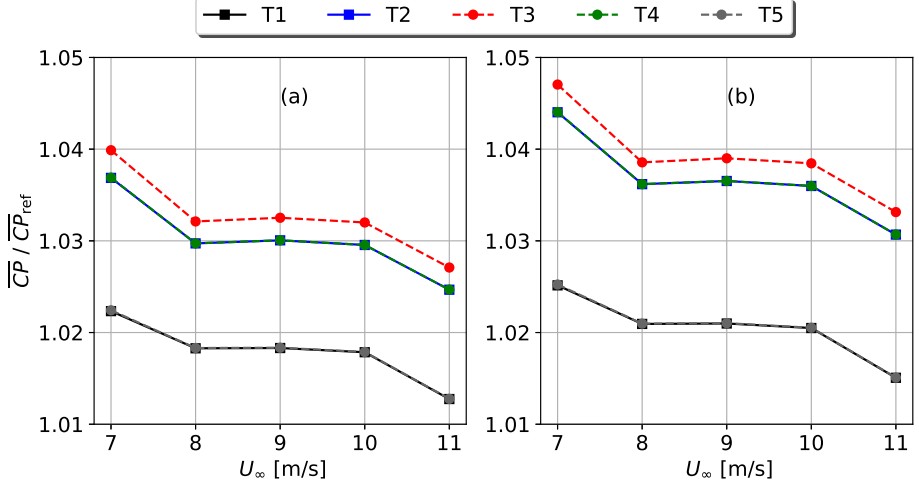

**Figure 9.** Power coefficient $C_P$ of the five turbines in the row averaged over the inflow sector and normalized by the average $C_P$ of the
isolated turbine for several values of $U_\infty$ and a normal distribution $N_\theta(\mu_\theta = 0°, \sigma_\theta = 41°)$ for $\theta$. The free-stream velocity $U_\infty$ is extracted
by a virtual met mast (*a*) and a virtual lidar (*b*) at a distance of 2.5D from the rotor.

## 4   Analysis of the measurements

### 4.1   Data filtering

To compute the power variation $P(\theta)$ observed at the site, the inflow sector $\theta = 0° \pm 50°$ is divided into three different intervals

$\theta = 0° \pm 16.5°$, $+33° \pm 16.5°$ and $-33° \pm 16.5°$. These are selected to characterize the three conditions of operation ("upwind",

"downwind" and inflow perpendicular to the row) and obtain the largest possible amount of data within each interval. The wind

direction is taken from measurements of a wind vane installed on the met mast 4 m below hub height. Additionally, the data

are filtered according to the wind speed measured by the hub-height cup anemometer ($U_{HH}$) and corrected for air density,

even though the correction leaves the data nearly unchanged due to the flat and sea-level terrain. Only wind speeds within

the constant $C_P$ range are considered, as this is also the range providing a constant $C_T$ (the manufacturer's $C_T$-curve is not

available). Therefore, after determining the $C_P$ curve of the turbine, the interval $U_{HH} = [5.5, 8.5]$ m/s is selected.

After selecting the data according to 10-min mean values of both $\theta$ and $U_{HH}$, other meteorological conditions are imposed

to both increase the compliance with the numerical setup and to avoid biases due to extreme conditions. Conditions of both



very low and very high turbulence are filtered out by considering only 10-min intervals where the turbulence intensity at hub height is between 2% and 10%. Additionally, thresholds are set for both the wind veer ($\gamma$) and the wind direction standard deviation ($\sigma_\theta$). Measurements with either $\gamma > 10°$ or $\sigma_\theta > 10°$ are filtered out. $\gamma$ is the difference between the 10-min-mean wind directions given by the WindCube at heights of $-0.4D$ and $0.85D$ relative to hub height.

We also consider only power-law-like wind profiles to avoid biases due to different profiles among different wind directions. For all the 10-min intervals, the power law is fitted to the measurements at the 11 different heights measured by the Wind-Cube. Then, the mean absolute error (MAE) between the measured wind speeds and the values estimated by the power law is calculated. Only the profiles providing a MAE lower than 0.03 are taken for the analysis to avoid reducing the amount of data excessively. Additionally, to avoid conditions of very strong shear, profiles with a shear exponent $\alpha$ higher than 0.35 are

discarded. Finally, to increase the amount of data, we select all the intervals when at least two of the other four turbines are operating.

### 4.2    Power variations

Due to the substantial differences between the numerical setup and the real site, the objective of the inter-comparison with the measurements is to evaluate the trends of power variations. Thus, we evaluate the power output for each of the three

wind direction bins ($\theta = 0° \pm 16.5°$, $\theta = +33° \pm 16.5°$ and $\theta = -33° \pm 16.5°$) and make sure that the same meteorological conditions are in place in all the three bins, so that the power differences are mainly explained by the effect of the other four turbines. However, we could have different wind speed distributions among the bins, since the wind speed interval is relatively large (3 m/s). Therefore, we normalize the 10-min mean power values $P_i$ with the power value derived from the power curve for the related 10-min mean wind speed measured at hub height, resulting in the normalized power values $\hat{P}$. The power

curve is derived from the dataset filtered for meteorological conditions, without including the operational status of the other turbines. Different atmospheric stability conditions might be associated with different wind directions. To decrease the effect of stability, data are sampled so that the three bins present the same number of measurements within each interval $\alpha = \bar{\alpha} \pm 0.02$, for $\bar{\alpha} = 0.01, 0.03, 0.05, .., 0.33, 0.35$. This sampling assures that all the inflow sectors have the same distribution of $\alpha$ values and it results in 534 10-min mean data for each of the 3 sectors. The distributions for the normalized power $\hat{P}$ are shown

in figure 10. Additionally, the sampling for $\alpha$ is repeated for 50 random seeds and the results are nearly constant (standard deviations lower than 0.1% of the means), proving that the findings are not affected by the random sampling.

The highest mean power output is observed for $\bar{\theta} = -33°$ ($\bar{\theta}$ stands for the mean of all the 10-min wind directions), i.e., when the turbine is the most downwind in the row. The lowest power output is observed for $\bar{\theta} = 32°$, when the wind turbine is the most upwind. The comparison between measurements and simulations is shown in figure 11, where the numerical results,

indicated as red squares, represent the means of the power distributions obtained for the same wind direction distribution of the measurements and a free-stream velocity of 8 m/s. To ease the comparison, as the simulations do not correspond to the same turbine; the means and uncertainties are normalized by the mean power output for the central sector ($\bar{\theta} = 1°$). As illustrated, both simulations and observations show the same trend. Also, the differences among the mean power values are larger than the



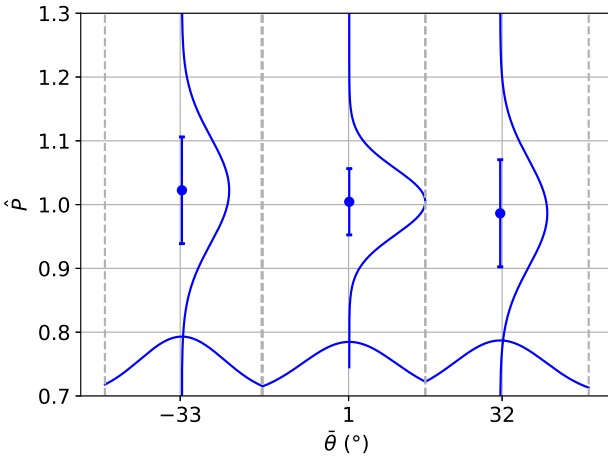

**Figure 10.** Power output normalized by the power derived from the power curve for different wind directions. Dots and error bars represent means and standard deviations within each bin, while the continuous lines represent the distributions of $\hat{P}$ and $\theta$ within each bin

uncertainties, which represent 95% confidence intervals. The statistical significance of the results is also tested through null
hypothesis significance testing, resulting in p-values below 0.05 for all the inflow angles.

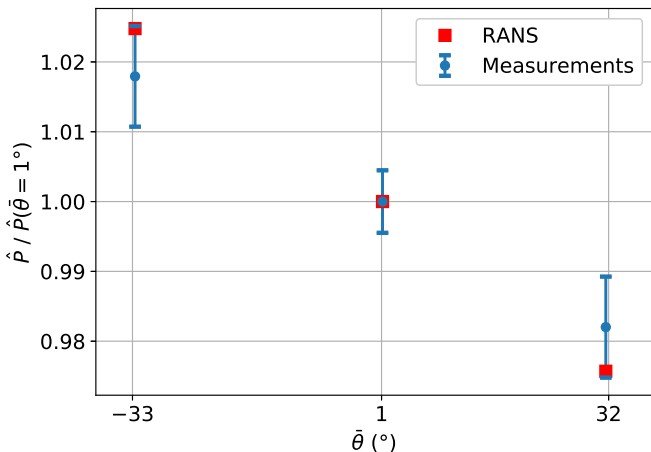

**Figure 11.** Normalized power output variation with wind direction based on measurements and simulations. Error bars represent 95% confidence intervals.



## 5 Discussion

The RANS simulations show that the power output of five turbines in a row is higher than what they would produce in isolation when the incoming wind is perpendicular to the line along the rotor. This is in agreement with the study of McTavish et al. (2015) for a line of three turbines, with that of Meyer Forsting et al. (2017b) for a row of five turbines and that of Van der Laan et al. (2019) for a multi-rotor configuration. Additionally, the RANS simulations show that the power difference between the reference and the five aligned turbines changes when the wind is not perpendicular to the row. Specifically, the downwind turbines produce more power than the upwind turbines, with a difference that increases for larger inflow angles. For the cases with $\theta = 30°$ and $45°$, the most upwind turbine produces less than the isolated turbine for all simulated turbine inter-spacings (1.8D, 2D and 3D). These results agree with those by Meyer Forsting et al. (2017b) for the case with $U_\infty = 8$ m/s and a turbine inter-spacing of 3D, despite the addition of a wind turbine controller. However, adding the controller results in a different outcome for the cases with $U_\infty = 7$ and 11 m/s, as we are not any longer within the constant-$C_P$ region of operation of the turbine.

The power variations are due to positive and negative velocities induced in the wakes of the neighbouring rotors, which depend on $\theta$. In this study, we show how these induced velocities vary with both the turbines inter-spacing and the free-stream velocity. Furthermore, wind profiles extracted upstream of the rotors show that the row's global blockage causes a reduction of the upstream velocity relative to the isolated case, as expected. However, it has the opposite effect on the power output, which might be counter-intuitive. When averaging over the whole inflow sector ($-45° < \theta < 45°$), an increase relative to the isolated case of more than 1% for the power and more than 4% for the power coefficient $C_P$ further confirms our results.

Analysis of field and SCADA measurements confirm the numerical results. Due to differences between the numerical setup and the conditions of the measurements, we cannot expect a one-to-one agreement between measurements and simulations. We expected the global-blockage effect at the site to be lower than in the simulations, since the turbine inter-spacing is larger at the site and since we also consider cases where either three, four or five turbines are in operation. Nevertheless, measurements show a very good agreement with the numerical results. This might be due to an increase of the global-blockage effect due to wind shear and terrain effects. The terrain represents an additional boundary to the flow, deflecting greater amounts to the sides and above the wind farm. This effect, usually simulated with mirror rotors (Meyer Forsting et al., 2021), is not accounted for in our setup and causes an additional source of blockage in the real site. In sheared inflow conditions, the wind speed in the lower part of the rotor can be substantially lower than that in the higher part, which causes a non-optimal selection of the pitch angle for the blades and consequently a non optimal aerodynamic performance of the wind turbine. As shown by Meyer Forsting et al. (2018), the blade forces in the lower half of the rotor are stronger than what would be the optimal value according to the local velocity. This results in a higher local $C_T$ and stronger induction in the lower half of the rotors.

The measurements confirmed the relation between power and wind direction $P(\theta)$, with an increased power output when the side turbine is the most downwind and a decreased power output when it is the most upwind relative to the case for $\theta = 0°$. The filtering procedures applied to the measurements try to guarantee that the power variation $P(\theta)$ is driven by blockage effects as it is the case in the simulations.





It must be noted that the measurements analyzed in this work cover only the constant-$C_P$ region of the power curve, so only a small portion of the power curve is subjected to the power increase. Numerical results show that the power rise is higher for wind speeds below that region, while it is lower, although still present, for wind speeds above it.

## 6 Conclusions

The power output of five wind turbines in a row is computed through RANS simulations and compared with the power output of the same turbine in isolation. The flow field is also analyzed both upstream and downstream to understand the global-blockage effect resulting from the wind-farm orientation. Several cases are considered, with variations to the free-stream velocity, the turbine inter-spacing and the inflow angle.

Our results show that the power output varies according to all the above three factors, with changes relative to the isolated case from $-3\%$ to $+5\%$. We find an increase of more than $1\%$ for the mean power output when averaging over the whole inflow angle ($-45° < \theta < 45°$) for a turbine inter-spacing of 2D and several values of the free-stream velocity. Due to the upstream velocity reduction caused by global blockage, the difference with the reference increases up to $4\%$ for the mean power coefficient.

Measurements from a site are analyzed in order to validate the numerical findings. The site consists of five turbines in a row and the available dataset comprises the operational data from one of the side turbines together with measurements from both a met mast and a ground-based WindCube lidar located in front of the turbine. The analysis confirms the variation of power with inflow angle observed in the simulations. Compared to the case with a flow perpendicular to the row, the power output changes of $+(1.8 \pm 0.7)\%$ and $-(1.8 \pm 0.7)\%$ when the turbine is the most downwind and upwind of the line, respectively.

Our work shows that wind turbine power output can be enhanced when wind turbines are aligned on a row. We also show how power performance testing might be biased when performed on such an array of wind turbines with an inter-spacing below 3D.

*Data availability.* Data from the turbines, the meteorological mast and the WindCube are not publicly available due to a non-disclosure agreement between the authors and the provider of the data.



*Author contributions.* AS, AP, NT and AMF participated in the conceptualization and design of the work. NT and AMF implemented the numerical setup for the CFD simulations. AP was responsible for the acquisition of the dataset. AS performed the CFD simulations, conducted the data analysis and wrote the draft manuscript. AP, NT and AMF supported the whole analysis and reviewed and edited the manuscript.

*Competing interests.* The authors declare that they have no conflict of interest.

*Acknowledgements.* We would like to thank Andrea Vignaroli for discussions on the analysis of the wind turbine measurements. This work has received funding from the European Union Horizon 2020 through the Innovation Training Network Marie Skłodowska-Curie Actions: Lidar Knowledge Europe (LIKE), grant number 858358.



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
