# Peer review of "Evaluation of the global-blockage effect on power performance through simulations and measurements"

_Wind Energy Science, 2021_

## Referee Comment (RC1)

**Review for "Evaluation of the global-blockage effect on power performance through simulations and measurements" by Alessandro Sebastiani et al.**

**Referee:** James Bleeg

**General comments**

I am not aware of any other field observations in the literature directly demonstrating how global blockage effects (as defined in the manuscript) can influence the outcome of a turbine power performance measurement. The CFD simulations presented in the paper offer a useful complement to the observations, helping to provide a physical understanding of the trends. While the wind energy community seems vaguely aware that the undue influence of blockage can distort the outcome of a power performance test, very little as of yet is being done about it. Research such as this can help move the conversation forward to action.

Although I do not believe that the manuscript requires any major changes, I nevertheless have some change requests. They generally relate to adding detail and clarification in the interpretation of the results—and contextualizing them. Please see the next section for the change requests.

**Main specific comments**

**Some key conclusions need to be softened or caveated.** Even though the paper focuses on power performance, it also states multiple times that the turbines in a row produce on average 1% more power than they would operating in isolation. The finding that interrow interactions can increase power production is consistent with findings from other referenced researchers as pointed out in the Discussion section (your references are fine, but if you wanted to you could also add Nishino and Draper "Local blockage effect for wind turbines" as well as Strickland and Stevens "Investigating blockage effects in Large-Scale…"). Based on the results in the manuscript, and probably with added confidence derived from similar findings from others, the conclusions state "our work shows that wind turbine power output can be enhanced when wind turbines are aligned on a row."

All the evidence I have seen for this conclusion comes from simulations and experiments involving neutrally stratified flow (i.e. zero buoyancy). At the same time, there is strong evidence in the literature indicating that stable stratification, both within and above the boundary layer, has a first-order impact on blockage effects (e.g Schneemann et al and Allaerts and Meyers from the reference list). My own work in this area indicates that once atmospheric stability is accounted for in the simulation, the single-row production gains generally go away or are reversed (https://winddenmark.dk/node/2042, first download on this page).

Another factor that may materially affect the turbine interaction gains reported in the paper is the lack of shear and ground effect in the simulations. When actuator disks are simulated above a ground surface, the streamlines generally rise as they approach the wind farm such that the height of a streamline passing through the rotor is generally higher than the height of the streamline far upstream. The height differences are on average more positive than what would occur for a simulation of an isolated turbine. The significance of this is when the simulation is also run with vertical shear, the flow passing through the rotor within a wind farm originates from a lower height—with a lower energy flux—than flow passing through the turbine were it to be operating in isolation. The impact on power probably is not large, but it very well may be on the order of the ~1% trends noted in the paper.

For these reasons, I think the conclusion regarding the enhancement of power output for turbines aligned in the row should be scaled back or clearly caveated. At the least, the manuscript should indicate that this conclusion pertains to pure neutral conditions only. Different trends may be found when accounting for atmospheric stability within and/or above the boundary layer. All that said, I actually agree that there are probably combinations of turbine spacing and atmospheric conditions that do enhance power production. My concern is that as currently written the paper could leave the reader with the impression that this is generally the case in the field, and I do not believe the evidence as it currently stands is strong enough to support such a conclusion.

**More could be done to explain the significance of the main conclusion**. In my view, the most important finding is the evidence indicating that "power performance might be biased when performed on such an array of wind turbines with an inter-spacing below 3D." This conclusion largely derives from the results summarized in figures 9-11. At the least, the measurements demonstrate that global blockage effects, as defined in the paper, can materially influence the outcome of a power performance test. The CFD results further indicate the measured performance of a turbine in a row of turbines could be  greater than what would be measured for a turbine operating in isolation—by around 2-4% for below-rated conditions in the simulated row. I think this what the authors are referring to when they say there is a bias, but I'm not sure.

I agree that the findings signify a bias, but I think more could be done to specify precisely what the bias is relative to. Is this bias limited to consideration of the outcome of a power performance test run on a turbine located in a row compared to a measurement with the turbine in isolation? Do the authors have a view as to whether there is also a bias relative to what is described on line 57 as the ideal definition of a power curve?

In addition, it might help the reader if these findings were put in context. One might argue that regardless of what would be measured for a turbine in isolation, the power curve measurement for the turbine in the wind farm is more relevant to consideration of how the turbine is performing in that wind farm. I would disagree with such an argument, but it would be good to hear the view of the authors.

My own view is as follows. There is no requirement to use it in the paper; I just want to provide an example of what I have in mind. When the gross energy is calculated during an energy yield analysis (EYA), the estimated freestream wind speed frequency distribution at hub height at each wind turbine location is fed directly into a power curve. Therefore, an accurate estimate of the gross energy  of each turbine requires a power curve that faithfully represent how much energy the turbine produces when operating in isolation for the given freestream wind resource. In my view, your results strongly suggests that global blockage effects can cause measured power curves to depart from the power curve that is needed in an EYA. Further, since measured power curves influence the curves used in EYAs (there is plenty of evidence indicate that measured curves are similar to and no more energetic than those used in EYAs), this signifies a bias in EYAs.

**Definition of power performance.** When the term "power performance" is used in this paper, I believe it always refers to measured power performance or simulated 'measured' power performance. In other words, when on line 15 you write, "we also show that the power performance is impacted by the neighboring turbines," you are referring to the measured power performance. Further, you are not saying that the actual power performance of the turbine is impacted by

neighboring turbines. Is my understanding correct? If my assumptions are correct, I think it would be a good idea to explicitly clarify this in the manuscript.

If my assumption is not correct, it would be to get a clearer understanding of your view on this.

**Minor specific comments**

- Line 85: I think parentheses should be around the two Meyer Forsting references.
- Line 135: Maybe this is a UK vs US thing, but I would replace ", which consists of" with "consisting of" (no comma)
- Line 223: How did you come up with a $C_T$ level for the simulated turbines? What is that level?

---

## Author Response (AR1)

**Response to referee 1**

Dear James Bleeg,

Thanks for your general comments on our work. We are glad that you think it is a valuable contribution to the wind energy community. We are also thankful for your specific comments that certainly contribute to increase the value of the manuscript. Please, find our answers below. Comments from the reviewer are reported in bold and followed by our answers.

Main specific comments

1. Some key conclusions need to be softened or caveated

**Even though the paper focuses on power performance, it also states multiple times that the turbines in a row produce on average 1% more power than they would operating in isolation. The finding that interrow interactions can increase power production is consistent with findings from other referenced researchers as pointed out in the Discussion section (your references are fine, but if you wanted to you could also add Nishino and Draper "Local blockage effect for wind turbines" as well as Strickland and Stevens "Investigating blockage effects in Large-Scale…"). Based on the results in the manuscript, and probably with added confidence derived from similar findings from others, the conclusions state "our work shows that wind turbine power output can be enhanced when wind turbines are aligned on a row."**

Thank you for suggesting additional literature. We consider both works as strongly related to ours and we added them in the Introduction section of the revised manuscript.

**All the evidence I have seen for this conclusion comes from simulations and experiments involving neutrally stratified flow (i.e. zero buoyancy). At the same time, there is strong evidence in the literature indicating that stable stratification, both within and above the boundary layer, has a first-order impact on blockage effects (e.g Schneemann et al and Allaerts and Meyers from the reference list). My own work in this area indicates that once atmospheric stability is accounted for in the simulation, the single-row production gains generally go away or are reversed (https://winddenmark.dk/node/2042, first download on this page).**

Although stability and other atmospheric effects can affect global blockage, in our work we disregard the impact of the atmosphere, thus we only consider the mutual interaction of the turbines. This simplification allows us to direct our attention in one of the mechanisms behind blockage effects. It should be also noted that the relation between global blockage and atmospheric conditions was studied for infinite or at least very large wind farms (Allaerts & Meyers, 2017; Porté-Agel et al., 2020; Schneemann et al., 2021), while the mentioned reviewer's work (https://winddenmark.dk/node/2042) consists of a row of 21 wind turbines. We neglect the impact of the atmosphere, as we do not expect that gravity waves can be induced in the case of a single row of five wind turbines. Additionally, when accounting for atmospheric conditions, results are more likely to depend on the modelling choices.

**Another factor that may materially affect the turbine interaction gains reported in the paper is the lack of shear and ground effect in the simulations. When actuator disks are simulated above a ground surface, the streamlines generally rise as they approach the wind farm such that the height of a streamline passing through the rotor is generally higher than the height of the streamline far upstream. The height differences are on average more positive than what would occur for a simulation of an isolated turbine. The significance of this is when the simulation is also run with vertical shear, the flow passing through the rotor within a wind farm originates from a lower height— with a lower energy flux—than flow passing through the turbine were it to be operating in isolation. The impact on power probably is not large, but it very well may be on the order of the ~1% trends noted in the paper.**

Although we know that global blockage can be affected by shear and ground effects, we disregard these effects in the simulations, as for the atmospheric stability, in order to focus on the effects related to the mutual interactions between the rotors. Additionally, previous studies found a similar power enhancement for a single row of wind turbines when including ground and shear with a neutral atmospheric boundary layer (McTavish et al., 2015; Strickland & Stevens, 2020). Based on the latter studies, we would expect that the power enhancement shown in our work would not vanish but rather increase when adding both ground and shear effects.

**For these reasons, I think the conclusion regarding the enhancement of power output for turbines aligned in the row should be scaled back or clearly caveated. At the least, the manuscript should indicate that this conclusion pertains to pure neutral conditions only. Different trends may be found when accounting for atmospheric stability within and/or above the boundary layer. All that said, I actually agree that there are probably combinations of turbine spacing and atmospheric conditions that do enhance power production. My concern is that as currently written the paper could leave the reader with the impression that this is generally the case in the field, and I do not believe the evidence as it currently stands is strong enough to support such a conclusion.**

We agree that the numerical results presented in our work are representative of only specific conditions due to the lack of shear, buoyancy and ground in the simulations. Therefore, to ensure the highest possible similarity between the numerical setup and the conditions of the measurements, we applied the filtering and binning described in sections 4.1 and 4.2. Specifically, we filtered out all the wind profiles that do not follow a power-law-like profile (e.g. wind profiles presenting low-level jets) and all the wind profiles presenting a very strong shear (power law exponent $\alpha$ higher than 0.35). This means that all the power-law-like wind profiles presenting $0 \leq \alpha \leq 0.35$ are considered despite of the atmospheric stratification. To make sure that the observed power differences are not due to different profiles among different wind directions, we make sure that each wind direction bin presents the same distribution of $\alpha$, as described in section 4.1. However, despite of the filtering, we did not expect a one-to-one agreement between measurements and simulations, because both the turbine inter-spacing is larger at the site and we also consider cases where either three, four or five turbines are in operation. As mentioned in the Discussion (lines 281-290), the surprisingly good agreement might be due to an

increase of the global-blockage effect at the site due to wind shear and terrain effects, which are missing in the numerical setup as pointed out by the reviewer.

For what concerns atmospheric stability, we did not focus on it in our work and it might be something to take into account for further extension of this research. It would be interesting to evaluate how the global-blockage effect in a single row of wind turbines changes for different atmospheric conditions. However, because of the wind climatology of the site, we believe that most of the measurements are collected under near-neutral conditions and this might be an additional reason for the good agreement between measurements and simulations.

We agree with the reviewer that the evidence we provide for the enhancement of power output is not such to state that the power output of wind turbines aligned in a row is always higher than that of an isolated turbine operating under the same conditions. Therefore, as suggested by the reviewer, to prevent the readers from assuming our results as proof of a general condition, in the revised version we will make sure to mention in both Abstract and Conclusions that the simulations pertain to pure neutral conditions with no shear and no ground. Nevertheless, it should also be mentioned that the good agreement between the measurements and the simulations show that, even under such simplifications, the numerical results represent well the interactions between wind turbine rotors in the real world.

2. More could be done to explain the significance of the main conclusion.

**In my view, the most important finding is the evidence indicating that "power performance might be biased when performed on such an array of wind turbines with an inter-spacing below 3D." This conclusion largely derives from the results summarized in figures 9-11. At the least, the measurements demonstrate that global blockage effects, as defined in the paper, can materially influence the outcome of a power performance test. The CFD results further indicate the measured performance of a turbine in a row of turbines could be greater than what would be measured for a turbine operating in isolation—by around 2-4% for below-rated conditions in the simulated row. I think this what the authors are referring to when they say there is a bias, but I'm not sure.**

When stating that "power performance testing might be biased", we refer to both the power increase shown by the numerical results and the influence of global blockage demonstrated by the measurements, as suggested by the reviewer.

The numerical results (figures 8 and 9) show that both the power output and the power coefficient $C_P$ are enhanced (under neutral conditions and no effects from both shear and ground) when five wind turbines are aligned on a row, with an increase of more than 1% in power output and more than 4% in $C_P$ compared to an isolated turbine operating under the same conditions. Therefore, this might cause a bias for power performance measurements conducted on a row of wind turbines, as they would result in a better power performance than that measured for the same turbine in isolation. Additionally, the measurements show that global-blockage effects, as defined in this work, influence the power output of a row of wind turbines causing possible biases for power performance tests conducted on such array. As shown in figure 11, the power output changes with wind direction due to the global-blockage effect. This means that a different power output could be obtained from the same turbine depending on the

wind direction distribution and the number of turbines in the row, resulting therefore in a bias for the power performance measurement, which is assumed as independent of those variables.

**I agree that the findings signify a bias, but I think more could be done to specify precisely what the bias is relative to. Is this bias limited to consideration of the outcome of a power performance test run on a turbine located in a row compared to a measurement with the turbine in isolation? Do the authors have a view as to whether there is also a bias relative to what is described on line 57 as the ideal definition of a power curve?**

The final statement is modified in the revised manuscript to increase clarity regarding the bias. Specifically, the numerical results show that power performance tests conducted on a single row of wind turbines might result in a better power performance than what would be measured for the same turbine in isolation. Additionally, the measurements show that, in a single row of wind turbines, the power output changes with the wind direction due to the global-blockage effect. This suggests that a different power output could be obtained depending on the wind direction distribution, resulting in possible biases for power performance tests conducted on such array.

As mentioned at line 57, power curves ideally define the relation between the wind turbine power output and the wind speed that would be measured at the turbine's location without the turbine actually being there. One could argue that the IEC standard procedure is seriously questioned by the evidence provided in this work for global blockage, when the procedure is performed in a single row of wind turbines. That is because the wind speed measured at a distance of 2D in front of the rotor would be affected by global blockage, causing a larger difference between the measured wind speed and the "ideal" wind speed than in the isolated case. However, evaluation of the global-blockage effect on the wind speed measurements in a single row of wind turbines is out of the paper scope and it is something to take into account for possible extensions of this work. Such evaluation could be certainly conducted numerically, e.g. through RANS or LES simulations. However, the main difficulties would be in validating the numerical results with measurements, as it would not be easy to measure the velocity variations with an uncertainty that is lower than the variations themselves. Additionally, one would ideally need data of the same turbine operating in both isolation and within a single row of wind turbines under similar conditions.

**In addition, it might help the reader if these findings were put in context. One might argue that regardless of what would be measured for a turbine in isolation, the power curve measurement for the turbine in the wind farm is more relevant to consideration of how the turbine is performing in that wind farm. I would disagree with such an argument, but it would be good to hear the view of the authors.**
**My own view is as follows. There is no requirement to use it in the paper; I just want to provide an example of what I have in mind. When the gross energy is calculated during an energy yield analysis (EYA), the estimated freestream wind speed frequency distribution at hub height at each wind turbine location is fed directly into a power curve. Therefore, an accurate estimate of the gross energy of each turbine requires a power curve that faithfully represent how much energy the turbine produces when operating in isolation for the given freestream wind resource. In my view, your results strongly suggests that global blockage effects can cause measured power curves to depart from the power**

**curve that is needed in an EYA. Further, since measured power curves influence the curves used in EYAs (there is plenty of evidence indicate that measured curves are similar to and no more energetic than those used in EYAs), this signifies a bias in EYAs.**

Our own view is that IEC standard for power performance measurements should ideally provide an accurate representation of the power curve of the wind turbine in isolation. However, to our knowledge, a number of power performance tests are carried out in a non-isolated situation, thus resulting in power curves that are different from those for isolated turbines. As we show, biases in power depend, among others, on both the wind direction and the $C_T$ of all the turbines in the row. This suggests that power performance tests do not result in a general "wind farm power curve" when carried out in a non-isolated situation. Therefore, due to the lack of a validated methodology to assess the power performance of a wind turbine inside a wind farm, we should try to minimize the influence from other turbines to get a generally valid power curve.

This consideration is now added in the Discussion section of the revised manuscript.

3. Definition of power performance

**When the term "power performance" is used in this paper, I believe it always refers to measured power performance or simulated 'measured' power performance. In other words, when on line 15 you write, "we also show that the power performance is impacted by the neighboring turbines," you are referring to the measured power performance. Further, you are not saying that the actual power performance of the turbine is impacted by neighboring turbines. Is my understanding correct? If my assumptions are correct, I think it would be a good idea to explicitly clarify this in the manuscript. If my assumption is not correct, it would be to get a clearer understanding of your view on this.**

There is a difference between "power performance" and "measured power performance". With "power performance", we refer to how much the wind turbine produces under certain conditions, while "measured power performance" refers to the outcome of a power performance test, which consists in measuring the power output and the wind speed at the same time and location to evaluate their relation. Our results, from both simulations and measurements, show that global blockage affects the power performance of wind turbines aligned in a single row (figures 3, 4, 10, 11). Additionally, simulations show that global blockage affects power output and upstream velocity differently, i.e. it affects the relation between power output and incoming wind speed (figures 5, 8, 9). These results strongly suggest that power performance tests might be biased when performed on a row of aligned wind turbines. This difference is now clearer in both Discussion and Conclusions of the revised manuscript.

Minor specific comments

**Line 85: I think parentheses should be around the two Meyer Forsting references**.
The text is now modified according to the comment.

**Line 135: Maybe this is a UK vs US thing, but I would replace ", which consists of" with "consisting of" (no comma)**
The text is now modified according to the comment.

**Line 223: How did you come up with a $C_T$ level for the simulated turbines? What is that level?**

We do not define the $C_T$ curve of the simulated wind turbine. As specified in line 155, we know that for wind speeds between 8 m/s and 10 m/s "the turbine is within the region of the power curve where the turbine controller keeps a constant tip speed ratio and an optimal power output, i.e. a constant power coefficient ($C_P$) and thrust coefficient ($C_T$). " This information is derived from the technical report from the developers of the NREL 5MW wind turbine [*Jonkman, J., Butterfield, S., Musial, W., and Scott, G.: Definition of a 5-MW Reference Wind Turbine for Offshore System Development, https://doi.org/10.2172/947422, 2009*].

**References**

Allaerts, D., & Meyers, J. (2017). *Boundary-layer development and gravity waves in conventionally neutral wind farms*. 95–130. https://doi.org/10.1017/jfm.2017.11

Porté-Agel, F., Bastankhah, M., & Shamsoddin, S. (2020). Wind-Turbine and Wind-Farm Flows: A Review. In *Boundary-Layer Meteorology* (Vol. 174, Issue 1). Springer Netherlands. https://doi.org/10.1007/s10546-019-00473-0

S. McTavish, S. Rodrigue, D. Feszty, F. N. (2015). An investigation of in-field blockage effects in closely spaced lateral wind farm configurations : In-field blockage effects in closely spaced lateral configurations. *Wind Energy*, *September 2014*, 1–20. https://doi.org/10.1002/we

Schneemann, J., Theuer, F., Rott, A., Dörenkämper, M., & Kühn, M. (2021). Offshore wind farm global blockage measured with scanning lidar. *Wind Energ. Sci*, *6*, 521–538. https://doi.org/10.5194/wes-6-521-2021

Strickland, J. M. I., & Stevens, R. J. A. M. (2020). Effect of thrust coefficient on the flow blockage effects in closely-spaced spanwise-infinite turbine arrays. *Journal of Physics: Conference Series*, *1618*(6). https://doi.org/10.1088/1742-6596/1618/6/062069

**Response to referee 2**

Dear referee,

Thanks for your general comments on our work. We are glad that you think it is a valuable contribution to the wind energy community. We are also thankful for your specific comments that certainly contribute to increase the value of the manuscript. Please, find our answers below. Comments from the reviewer are reported in bold and followed by our answers.

**Major Comments**

**1. Line 35: The physical mechanism that produces wind plant blockage is not yet well understood and there is still not an accepted mechanism. Some simulations suggest an adverse pressure gradient, but others don't. Also, the upstream reverse pressure gradient is very sensitive to atmospheric conditions and LES code.**

It is true that there are several factors, which govern the global blockage and atmospheric conditions are indeed one of them as shown in several previous studies (Allaerts & Meyers, 2017; Porté-Agel et al., 2020; Schneemann et al., 2021). Those studies show the impact of atmospheric stability on large wind farms, which potentially can generate gravity waves with consequent adverse pressure gradients and thus a wind speed reduction.

In our work, we disregard the impact of the atmosphere, thus we only consider the mutual interaction of the turbines. This simplification allows us to direct our attention in one of the mechanisms behind blockage effects. Although stability and other atmospheric effects can affect the results, our work shows that the part of the global blockage that is always present (i.e. the part induced only by the rotors on each other) does affect the performance of turbines when placed in a standard test site setup. Additionally, it should be noted that the relation between global blockage and atmospheric conditions was investigated for infinite or at least very large wind farms (Allaerts & Meyers, 2017; Porté-Agel et al., 2020; Schneemann et al., 2021). We neglect the impact of the atmosphere, as we do not expect that gravity waves can be induced in the case of a single row of five wind turbines.

**2. Section 2.2.1: The author mentions that having the wake outside the refined region of the mesh alters the power output by ~1%. This is the same order of magnitude as the differences in power production due to blockage. Can the increase in wind turbine power be in part due to having the induction zone and wake outside the refined region of the mesh? Were any sensitivity studies done for the choice of the simulation domain with regards to having the induction zone outside the refined region?**

The ~1% difference in power output is found for the case with a staggered configuration (sketch on the right of figure 2), which is therefore not used for this study and which we suggest to avoid for studies that require high accuracy.  As mentioned in lines 116-119, with the standard configuration (sketch on

the left of figure 2), the power output varies for a maximum of ~0.15% among the different turbine locations.

The sensitivity of the results to the extension of the refined area was evaluated by replicating some of the results from figure 3 with the same layout (standard configuration) and an enlarged refined area (from y = −4D to y = +4D). However, enlarging the refined area results in differences in the order of 0.3% for both the isolated turbine and the five turbines, so that no difference is found for the results of figure 3. Therefore, we assumed that there is no need to enlarge the refined area and that our results are reliable, at least when evaluated in terms of normalized power output. This information is now added in the revised manuscript.

**3. Section 3.2 and Section 4.2: The author suggests induced velocities in the wake region produce higher power production relative to an isolated turbine. The induced velocities in Figures 6 and 7 are in the order of 0.8% at y = 0D. It is unclear how these small induced velocities downstream result in an increase in power production, when at the same time hub-height wind speeds are smaller than in the isolated case. Also, might these induced velocities be a numerical artifact since turbine measurements do not evidence this increase in power production? This issue should be investigated further before drawing strong conclusions about enhanced turbine performance.**

We believe that a difference in velocity of ~0.8% is not negligible and it could definitely be enough to explain the power variations observed in figures 3 and 4. For example, assuming the same air density and power coefficient values, such a velocity difference can result in nearly 2.5% of power variation. Additionally, as noted in line 174, the hub-height wind speeds are smaller than in the isolated case when evaluated at more than 0.2D from the rotors.

We do not believe that the induced velocities are numerical artifacts, as it was shown in a previous study that this numerical setup guarantees results free of tunnel blockage due to either grid resolution or domain size. The work is cited at line 114 of the manuscript [*Meyer Forsting, A. and Troldborg, N.: The effect of blockage on power production for laterally aligned wind turbines, Journal of Physics: Conference Series, 625, https://doi.org/10.1088/1742-6596/625/1/012029, 2015*]

**4. The authors conclude that wind turbine power output can be enhanced when turbines are aligned in a row. This is a very strong statement that has many caveats. Measurements show the increase in turbine power is only for the downstream turbine in the row. And the simulations are highly idealized (neutral potential temperature, uniform wind speed, unconstrained flow, no turbulence). Please modify.**

Note that Referee 1 also commented on this statement.

" We agree with the reviewer that the evidence we provide for the enhancement of power output is not such to state that the power output of wind turbines aligned in a row is always higher than that of an isolated turbine operating under the same conditions. Therefore, as suggested by the reviewer, to prevent the readers from assuming our results as proof of a general condition, in the revised version we will make sure to mention in both Abstract and  Conclusions that the simulations pertain to pure neutral conditions with no shear and no ground. Nevertheless, it should also be mentioned that the good

agreement between the measurements and the simulations show that, even under such simplifications, the numerical results are representative of the phenomena involved with the interactions between wind turbine rotors in the real world ".

Minor Comments

**- Section 1: improve literature review in paragraph 1.**

We now introduce one more study related to gravity waves and pressure gradients in large wind farms (Smith, 2010). Additionally, we add two more studies suggested by reviewer 1 (Nishino & Draper, 2015; Strickland & Stevens, 2020).

**- Line 60: It would be nice to have a number/order of magnitude to give more relevance to this work.**

Numbers are added in the revised version of the manuscript.

**- Line 89: are you considering a neutral potential temperature profile? Please specify.**

The inflow is purely neutral (no buoyancy). This information is now added in the revised version along the previous lines 89-91.

**- Section 2.2: I understand there is no need to describe the simulation setup in great detail, but it should be noted that there is no bottom boundary for the surface and thus the flow is completely unconstrained.**

We agree that this is an important information that should be stated clearly. We now add this information in section 2.2 of the revised manuscript.

**- Line 116: Clarify that you also want to test how the location of the turbine within the domain may influence the results.**

Clarification is now added in the revised version along line 118.

**- Line 167: Clarify. Based on Figure 3, it seems the largest variation in turbine power for T5 is for $theta$ = 45 deg rather than $\theta$ = 0 deg.**

Yes, this was a typo and it should be theta = 45$^o$.  This is now corrected in the revised version of the manuscript.

**- Line 174: In many numerical codes, the GAD parameterization distributes the forces over multiple grid cells along the streamwise direction. Are these grid cells contained in the region where the incoming wind speed for T3 is higher than in the isolated case (<0.2D)?**

In the utilized implementation of the actuator shape (P. E. Réthoré & Sørensen, 2012; Pierre Elouan Réthoré et al., 2014), the actuator-disk forces are smeared only over the direct neighbouring cells and translated into equivalent pressure jump at the cell faces. The refined inner box of the domain presents 128 equally spaced cells for a distance of 4D along the streamwise directon. This means that each cell covers a distance of around 0.03D and that most of the speed-up region (<0.2D) is not contained within the force distribution.

**- Section 3.3: Why a standard deviation of 41 deg?**

The value of 41⁰ was chosen in order to get a nearly uniform distribution of wind directions within the interval [-45⁰, 45⁰]. Although it is not shown in the paper, a narrower Gaussian distribution would enhance the increase of power for T3 and the results would not be representative of power performance tests in general, but rather of tests conducted with that specific and narrow wind direction distribution.

This information is added in the revised version of the manuscript.

**- Line 230: Please include the equation for the power-law wind profile for completeness.**

The equation is now included in the revised version of the manuscript

**- Line 258: Please clarify if the differences among the mean power values is between measurements and simulations, or between the different inflow angles.**

It is the difference between the mean values provided by the measurements for the different inflow angles. We now clarify this in the revised version of the manuscript.

**- Line 279: The author mentions SCADA measurements confirm the numerical results. It is unclear which numerical results since you mention the induced velocities in the paragraph immediately above, but the existence of these induced velocities is not confirmed by the measurements. Please clarify that measurements only confirm the overall trend in change in power when compared to a turbine in isolation.**

Yes, as suggested by the reviewer, this is now clarified in the revised version of the manuscript.

**- Figure 10: Confidence intervals rather than a standard deviation may provide more information to the reader.**

We use standard deviations in figure 10 in order to give more information about the variability within each bin. Additionally, 95% confidence intervals are shown in figure 11, where it can be noted that the differences between the mean values are larger than the uncertainties, pointing out the statistical significance of the results.

**References**

Allaerts, D., & Meyers, J. (2017). *Boundary-layer development and gravity waves in conventionally neutral wind farms*. 95–130. https://doi.org/10.1017/jfm.2017.11

Nishino, T., & Draper, S. (2015). Local blockage effect for wind turbines. *Journal of Physics: Conference Series*, *625*(1). https://doi.org/10.1088/1742-6596/625/1/012010

Porté-Agel, F., Bastankhah, M., & Shamsoddin, S. (2020). Wind-Turbine and Wind-Farm Flows: A Review. In *Boundary-Layer Meteorology* (Vol. 174, Issue 1). Springer Netherlands. https://doi.org/10.1007/s10546-019-00473-0

Réthoré, P. E., & Sørensen, N. N. (2012). A discrete force allocation algorithm for modelling wind turbines in computational fluid dynamics. *Wind Energy*, *15*(7), 915–926.

https://doi.org/10.1002/WE.525

Réthoré, Pierre Elouan, Van Der Laan, P., Troldborg, N., Zahle, F., & Sørensen, N. N. (2014). Verification and validation of an actuator disc model. *Wind Energy*, *17*(6), 919–937. https://doi.org/10.1002/WE.1607

Schneemann, J., Theuer, F., Rott, A., Dörenkämper, M., & Kühn, M. (2021). Offshore wind farm global blockage measured with scanning lidar. *Wind Energ. Sci*, *6*, 521–538. https://doi.org/10.5194/wes-6-521-2021

Smith, R. B. (2010). Gravity wave effects on wind farm efficiency. *Wind Energy*, *December*, 1–20. https://doi.org/10.1002/we

Strickland, J. M. I., & Stevens, R. J. A. M. (2020). Effect of thrust coefficient on the flow blockage effects in closely-spaced spanwise-infinite turbine arrays. *Journal of Physics: Conference Series*, *1618*(6). https://doi.org/10.1088/1742-6596/1618/6/062069